# Prenatal Detection of a *FOXF1* Deletion in a Fetus with ACDMPV and Hydronephrosis

**DOI:** 10.3390/genes14030563

**Published:** 2023-02-23

**Authors:** Katarzyna Bzdęga, Anna Kutkowska-Kaźmierczak, Gail H. Deutsch, Izabela Plaskota, Marta Smyk, Magdalena Niemiec, Artur Barczyk, Ewa Obersztyn, Jan Modzelewski, Iwona Lipska, Paweł Stankiewicz, Marzena Gajecka, Małgorzata Rydzanicz, Rafał Płoski, Tomasz Szczapa, Justyna A. Karolak

**Affiliations:** 1Chair and Department of Genetics and Pharmaceutical Microbiology, Poznan University of Medical Sciences, 60-806 Poznan, Poland; 2Department of Medical Genetics, Institute of Mother and Child, 01-211 Warsaw, Poland; 3Department of Laboratory Medicine and Pathology, University of Washington School of Medicine, Seattle, WA 98105, USA; 41st Clinic of Obstetrics and Gynecology, Centre of Postgraduate Medical Education, 01-004 Warsaw, Poland; 5Department of Pathomorphology, Wolski Hospital, 01-211 Warsaw, Poland; 6Department of Molecular & Human Genetics, Baylor College of Medicine, Houston, TX 77030, USA; 7Institute of Human Genetics, Polish Academy of Sciences, 60-479 Poznan, Poland; 8Department of Medical Genetics, Medical University of Warsaw, 02-106 Warsaw, Poland; 9II Department of Neonatology, Neonatal Biophysical Monitoring and Cardiopulmonary Therapies Research Unit, Poznan University of Medical Science, 60-535 Poznan, Poland

**Keywords:** lethal lung developmental disorder, hydronephrosis, alveolar capillary dysplasia, 16q24.1, prenatal diagnosis, genome sequencing

## Abstract

Alveolar capillary dysplasia with misalignment of pulmonary veins (ACDMPV) is a lethal lung developmental disorder caused by the arrest of fetal lung formation, resulting in neonatal death due to acute respiratory failure and pulmonary arterial hypertension. Heterozygous single-nucleotide variants or copy-number variant (CNV) deletions involving the *FOXF1* gene and/or its lung-specific enhancer are found in the vast majority of ACDMPV patients. ACDMPV is often accompanied by extrapulmonary malformations, including the gastrointestinal, cardiac, or genitourinary systems. Thus far, most of the described ACDMPV patients have been diagnosed post mortem, based on histologic evaluation of the lung tissue and/or genetic testing. Here, we report a case of a prenatally detected de novo CNV deletion (~0.74 Mb) involving the *FOXF1* gene in a fetus with ACDMPV and hydronephrosis. Since ACDMPV is challenging to detect by ultrasound examination, the more widespread implementation of prenatal genetic testing can facilitate early diagnosis, improve appropriate genetic counselling, and further management.

## 1. Introduction

Alveolar capillary dysplasia with misalignment of pulmonary veins (ACDMPV, OMIM #265380) is a rare, lethal lung developmental disorder (LLDD) in neonates [1].

Histopathologically, ACDMPV is characterized by the presence of abnormal intrapulmonary shunt vessels (“misaligned pulmonary veins”) adjacent to the arteries, which show frequent marked thickened muscular walls [2,3]. The alveoli may be enlarged with thickened septa and few capillaries that are not positioned correctly within the wall of the alveolus [2]. 

Clinically, ACDMPV manifests with severe respiratory distress and pulmonary arterial hypertension refractory to therapy [2,4,5]. The first symptoms of ACDMPV usually occur within the first 24–48 h after birth, and newborns die within a few days to weeks after disease presentation [2,6]. Only a few cases of late milder ACDMPV manifestation and longer survival have been reported [7,8,9,10,11,12,13]. For patients with atypical presentation of ACDMPV, lung transplantation can be considered [11]. 

ACDMPV can be associated with extrapulmonary malformations involving cardiovascular or gastrointestinal systems [2,6]. Patients may also have urogenital anomalies of variable severity, including hydronephrosis [2,6]. 

Congenital hydronephrosis is characterized by significant dilatation of the renal pelvis, with subsequent urinary stasis caused by posterior urethral valves, vesicoureteral reflux, or obstruction at the level of the pelvic-ureteral and vesicoureteral junction [14,15]. While hydronephrosis may coexist with various genetic diseases, including ACDMPV, it is more frequently detected as an isolated condition [15]. Hydronephrosis occurs in 1–2% of pregnancies and is one of the most common defects detected during routine prenatal ultrasound evaluation in the second or third trimester [16]. In approximately 80% of cases, hydronephrosis is transient and resolves in early life with conservative management [16]. However, the likelihood of spontaneous resolution depends on the severity of the anomaly and, in some cases, surgical intervention is needed within the postnatal period [16]. 

Heterozygous single-nucleotide variants (SNVs) or copy-number variant (CNV) deletions involving *FOXF1* and/or its lung-specific enhancer at 16q24.1 have been detected in ~90% of cases with ACDMPV [4,17,18]. The pLI score of 0.96 indicates that it is almost completely intolerant to loss-of-function [19]. 

*FOXF1* encodes a forkhead-box family transcription factor [20] that plays a crucial role in the branching of lung tubular structures through the sonic hedgehog (SHH) signaling in epithelial cells [21,22]. Its expression in the lung is regulated by a distant lung-specific enhancer region located ~270 kb upstream to *FOXF1* [18]. This enhancer, along with the *FOXF1* promoter, is shared by *FOXF1* and the lncRNA gene *FENDRR* [23].

Although most genetic changes in ACDMPV arise de novo, a small fraction of *FOXF1* variants have been inherited from the mosaic mother [24]. Most of CNVs arise de novo on the maternal chromosome 16; to date, only five (~10%) CNV deletions have been reported to have arisen de novo on the paternal chromosome [13,18,25,26]. Of note, it has been speculated that loss of *FOXF1* and *FENDRR* on the paternal chromosome causes more severe cardiac defects, leading to fetal death or spontaneous miscarriage [25]. The involvement of *FOXF1* in the pathogenetics of ACDMPV has been demonstrated in mouse models, in which haploinsufficiency of *Foxf1* manifested in lung immaturity [27,28]. 

Thus far, a couple hundred cases of ACDMPV have been described worldwide. The vast majority of diagnoses have been made post mortem, based on histopathological evaluation and/or genetic testing, and only seven cases of ACDMPV have been detected prenatally [18,29,30,31,32,33]. Here, we present a patient with hydronephrosis and 16q24.1 CNV deletion identified in prenatal genetic testing, indicating the diagnosis of ACDMPV. 

## 2. Materials and Methods

### 2.1. Human Subjects 

Material was collected from the patient (amniocytes, peripheral blood, lung, and kidney tissue) and his parents (peripheral blood) after receiving informed consent in accordance with the Declaration of Helsinki. The study protocol was approved by the Ethics Committee at Poznan University of Medical Sciences.

### 2.2. Histopathological Evaluation

Histopathological evaluation was performed on slides from formalin-fixed paraffin-embedded kidney and lung tissue specimens obtained at autopsy.

### 2.3. Molecular Analyses

For invasive prenatal studies, DNA was extracted from amniocytes using the Sherlock AX DNA isolation kit (A&A Biotechnology, Gdansk, Poland), according to manufacturer’s instructions. Array comparative genomic hybridization (aCGH) in the fetus was performed using the 60K CytoSure Constitutional v3 microarray (Oxford Gene Technology, Oxford, UK). 

For further genetic testing, DNA was extracted from the peripheral blood of the newborn and his parents using Gentra Purgene Blood Kit (Qiagen, Germantown, MD, USA). Parental and proband DNA samples were tested for the presence of the CNV deletion using junction-specific PCR with DreamTaq DNA Polymerase (Thermo Scientific, Waltham, MA, USA), followed by Sanger sequencing to map the deletion breakpoints. To determine the parental origin of the detected CNV deletion, trio-based genome sequencing (GS) was performed using NEBNext^®^ Ultra™ II FS DNA Library Prep Kit for Illumina (New England BioLabs, Inc. Ipswich, MA, USA) and paired-end sequenced (2 × 150 bp) on NovaSeq 6000 (Illumina, San Diego, CA, USA). The parental origin of the observed chromosomal abnormality was determined by analyzing the informative single-nucleotide polymorphisms (SNPs) within the deletion region.

## 3. Results

The male proband was the third child of non-consanguineous Caucasian parents with no familial history of ACDMPV, hydronephrosis, or other anomalies. 

The non-invasive serum screening test revealed a higher risk of trisomy 21 (1:108) with pregnancy-associated plasma protein A (PAPP-A) of 0.71 MoM, free β human chorionic gonadotropin (free β-hCG) of 2.28 MoM, crown rump length (CRL) of 71.5 mm, and nuchal translucency (NT) of 2.4 mm (<95th percentile). A second-trimester ultrasound performed at 20 weeks of gestation revealed bilateral pyelectasis, and amniocentesis was performed for molecular analysis. 

A ~0.74 Mb CNV deletion encompassing *FOXF1* (OMIM #601089), *FOXC2* (OMIM #602402), *FOXL1* (OMIM #603252), *FENDRR* (OMIM #614975), *MTHFSD* (OMIM #616820), *LINC01081* (OMIM #614977), and *LINC01082* (OMIM #614977) was identified prenatally at 20 weeks of gestation, using aCGH and confirmed by GS (Figure 1A). The proximal and distal breakpoints of the deletion map at chr16:85,952,700/85,952,726 (hg38), within *Alu*Sz and chr16:86,694,061/86,694,087 (hg38) within *Alu*Y, respectively (Appendix A). The exact position of breakpoints is unknown due to 26 bp microhomology at the deletion junction site. Analysis of parental samples and the informative polymorphic markers showed that the deletion arose de novo on maternal chromosome 16 (Appendix A). Given the pathogenic nature of the identified 16q24.1 CNV deletion, the family was counselled regarding a suspected diagnosis of ACDMPV and poor prognosis.

The fetus developed hydrops and placental hypertrophy at 28 weeks of gestation. Due to anhydramnios and the risk of developing mirror syndrome, labor induction was performed at 34 weeks of gestation. The child was born with a weight of 4000 g, length of 48 cm, and Apgar score of 1. The newborn had major respiratory distress consistent with ACDMPV, and due to the irreversible nature of this disease, the family decided to accompany the child in palliative care. The child passed away within the first hour of life with comfort care, without intensive resuscitation.

Postmortem examination demonstrated a premature male infant with a right foot contracture, generalized edema, and fluid in the pleural and peritoneal cavities. Kidney evaluation showed bilateral pelvicalyceal dilatation and multiple cortical renal cysts, which are characteristic of renal dysplasia. Microscopic evaluation of the lungs showed immature lung parenchyma, arrested in the late canalicular stage of lung development, with abnormal thin-walled shunt vessels (“misaligned pulmonary veins”) accompanying hypertrophic pulmonary arteries in the same adventitial sheath. Diminished capillaries in the alveolar septa were also present. Overall, histopathological findings were consistent with the spectrum of ACDMPV. 

## 4. Discussion

In most cases, ACDMPV is first considered at birth, based on respiratory failure and pulmonary arterial hypertension [2]. However, similar clinical symptoms can also be related to other conditions, including those from the LLDD spectrum or idiopathic pulmonary arterial hypertension [6]. Thus, differential diagnosis requires confirmation by the detection of characteristic histopathological features in lung tissue, considered as a gold standard in ACDMPV diagnostics [1,2], or genetic testing. Because affected neonates are usually unstable for open lung biopsy, most ACDMPV diagnoses are made post mortem during lung autopsy [2]. 

With increasing availability, molecular testing is becoming an important tool for identification of ACDMPV-related abnormalities [6]; however, it is usually time consuming (may take several weeks and newborns die before a diagnosis is confirmed). Thus, we recommend rapid genetic testing that would allow for earlier diagnosis (days rather than weeks) and influence the decision-making process in critically ill infants. 

While routine ultrasound pregnancy screening enables early identification of severe fetal malformations, ACDMPV lung abnormalities are mainly undetectable from intrauterine imaging. Thus, ACDMPV is rarely suspected prenatally. To date, only seven patients with prenatally-detected ACDMPV (Figure 1B) have been reported, including six with 16q24.1 CNV deletion [18,29,30,31,32,33]. Due to the severity of the malformations, infants passed away soon after birth, or the parents elected to terminate the pregnancy after genetic confirmation of ACDMPV [18,29,30,31,32,33]. 

The first prenatal deletion involving *FOXF1* at 16q24.1 was identified in a fetus with cystic hygroma, a single umbilical artery, and fetal hydrops [29]. Unfortunately, histopathological examination of the lungs was not performed [29]. In a patient with prenatally detected pericentric inversion of chromosome 16, ACDMPV with atrioventricular septal defect (AVSD) and intestinal arthrodesis was confirmed after birth [30]. In another patient with CNV deletion at 16q24.1 identified in prenatal screening, autopsy examination confirmed ACDMPV with AVSD and bilateral superior vena cava [31]. Puisney-Dakhli et al. described a fetus with a suspected single ventricular congenital heart malformation in whom prenatal testing identified a CNV deletion at 16q24.1, and post mortem evaluation confirmed hypoplastic left heart syndrome and ACDMPV [32]. Another patient with prenatally identified CNV deletion at 16q24.1q24.2 had esophageal dilation, kidney malformation, lymphedema, AVSD, ventricular septal defect, and other abnormalities within the cardiovascular system [33]. Two fetuses with hydronephrosis associated with ACDMPV, caused by CNV deletions involving *FOXF1* and its enhancer, have also been described [18].

Here, we present another ACDMPV fetus with hydronephrosis seen on a prenatal ultrasound screening (20th week of gestation) in whom, due to this finding and a higher risk of trisomy 21 revealed in PAPP-A test earlier in pregnancy, invasive prenatal genetic screening was pursued. The de novo ~0.74 Mb *Alu-Alu* mediated CNV deletion involving *FOXF1* on maternal chromosome 16 has been detected. Of note, the majority of de novo ACDMPV deletions at 16q24.1 are *Alu*-mediated [34]. Based on the molecular findings and the presence of extrapulmonary anomaly often associated with ACDMPV, the mother was counselled regarding a suspected diagnosis of ACDMPV and about the postnatal clinical course, with poor prognosis. The clinical symptoms of the disease appeared after the child’s birth. The initial diagnosis of ACDMPV was confirmed post mortem, based on characteristic histopathological lung findings.

This case informs that detection of hydronephrosis in routine prenatal screening should prompt the physician to consider ACDMPV in differential diagnoses. A more detailed examination of gastrointestinal and cardiovascular systems, e.g., heart echo later in pregnancy, can be performed to screen for anomalies often associated with ACDMPV. 

While the most commonly applied approaches for prenatal screening are karyotype or microarray tests, GS is now increasingly being utilized [35]. The use of next-generation sequencing (NGS) as a tool in prenatal testing allows for rapid and effective detection of various molecular defects, including both CNVs and SNVs [36]. The major advantage of GS is its potential to detect variants in protein-coding genes, as well as non-coding regions of the genome, which improves the diagnostic yield of genetic testing and allows identifying molecular causes of rare congenital fetal disorders with complex inheritance, including LLDD. NGS-based prenatal testing can enable early disease diagnosis, which in turn may improve counseling for parents and influence further management and goals of care [37].

In summary, we present the first Polish patient with ACDMPV and hydronephrosis, in which prenatal genetic testing revealed a de novo CNV deletion at 16q24.1, involving *FOXF1*. Since ACDMPV is very challenging to detect by ultrasound examination, the more widespread implementation of prenatal genetic testing is warranted to facilitate early diagnosis, allow accurate counselling, and provide appropriate medical management.

## Figures and Tables

**Figure 1 genes-14-00563-f001:**
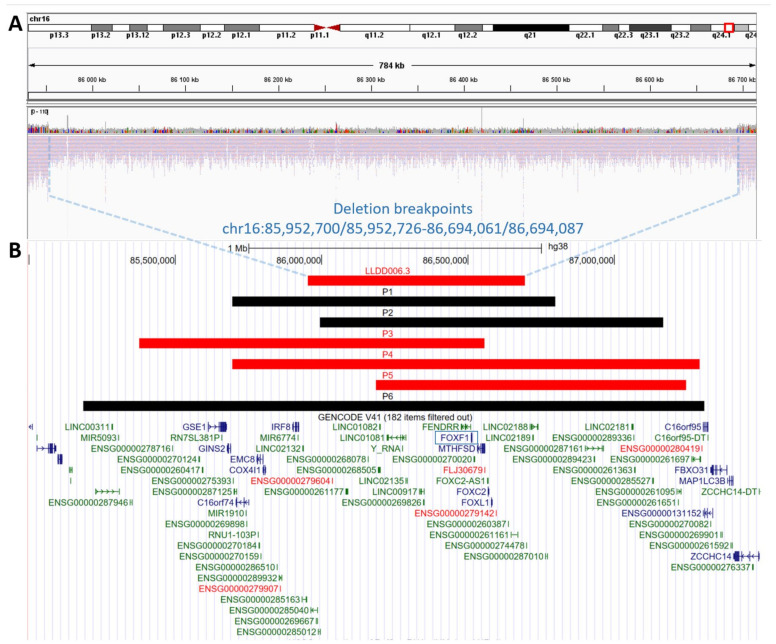
Schematic representation of the copy-number variant (CNV) deletions at 16q24.1 locus. (**A**) Genome sequencing display on Integrated Genome Viewer (IGV) showing a heterozygous ~0.74 Mb CNV deletion (chr16:85,952,700/85,952,726-86,694,061/86,694,087, hg38) in the assessed patient (LLDD006.3). (**B**) Comparison of ACDMPV-related CNV deletion identified in our patient during prenatal testing (LLDD006.3) with six previously described patients with prenatal ACDMPV diagnosis and 16q24.1 CNV deletion (P1 [29], P2 [31], P3 [32], P4 [18], P5 [18], P6 [33]). CNV deletions that arose on the maternal chromosome 16 are marked in red; CNV deletions with unknown parental origin are marked in black. The *FOXF1* gene is marked by a blue frame.

## Data Availability

The copy-number variant data were deposited in the dbVar database (https://www.ncbi.nlm.nih.gov/dbvar, accession number nstd222, accessed on November 2022).

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
