# Peer review of "Prenatal Detection of a *FOXF1* Deletion in a Fetus with ACDMPV and Hydronephrosis"

_genes, 2023, doi:10.3390/genes14030563_

Round 1
Reviewer 1 Report
Reviewer’s Comments:
Bzdega et al. describe a case of a newborn with alveolar capillary dysplasia with misalignment of pulmonary veins (ACDMPV) and hydronephrosis in whom a de novo 741 kb deletion of at 16q24.1 was identified prenatally. A deleted region harbors FOXF1, FOXC2, FOXL1, FENDRR, MTHFSD, LINC01081 and LIN01082. The authors claim they describe for the first time an association of ACDMPV –causative CNV deletion recognized prenatally in a fetus with hydronephrosis in Poland. They also determined the maternal origin of a deleted region of CNV deletion, using trio-based genome sequencing.
The manuscript adds to the literature. However, it needs elucidation and work.
1. Abstract
Page 1, line 29
The authors should not use the abbreviation “LLDD” for the first time without explanation and change it to “lethal lung developmental disorder”
2. “Since LLDDs, including ACDMPV are challenging to detect..”
I suggest writing: since ACDMPV, a lethal lung developmental disorder is challenging to detect by ultrasound examination.
3. Introduction
Since the authors are focusing on ACDMPV, I suggest starting the introduction with it, rather than hydronephrosis
4. Page 2, line 60
There have been more cases of ACDMPV reported prenatally.
The authors should also mention PMID: 36329475, PMID: 27071622
5. Page 3 Line 100
The authors should describe dysmorphic facial features
6. Since CNV deletion encompasses FOXC2, did the neonate have lymphedema?
7. Page 3 Line 109
They should mention OMIM genes and not disease-associated genes.
8. They should mention pL1 of FOXF1
9. Page 4 Line 145
There are more prenatally diagnosed cases of ACDMPV
10. Page5 Line 185
“Since LLDDs are very challenging to detect” change to ACDMPV
Reviewer 2 Report
In 'Prenatal detection of a FOXF1 deletion in a fetus with ACDMPV and 2 hydronephrosis' Bzdega and colleagues describe an infant who was prenatally identified to have a bilateral pyelectasis and amniocentesis and aCGH at 20 weeks gestation detected a 0.74Mb deletion including FOXF1. The report is concise and well-written.
Minor Comments
1. The authors should emphasize that pyelectasis and hydronephrosis, relatively common prenatal ultrasound findings, are generally isolated and not associated with underlying syndromes or anomalies of other organ systems. Many obstetricians/ fetal care centers would not necessarily recommend amniocentesis/ genetic testing for these findings.
2. ACDMPV has been identified in ethnically diverse patients throughout the world. Recommend not focusing on the specific country of birth for this infant.
3. While most patients with ACDMPV were previously diagnosed on autopsy or after death, infants are increasing being diagnosed during the NICU stay with genetic testing- recommend clarifying historic and current diagnostic context in text.
4. Was resuscitation attempted for this infant or did parents elect comfort care based on the aCGH findings? Recommend providing additional details for postnatal course.
5. Recommend reviewing references and including only the most relevant.
Round 2
Reviewer 1 Report
The authors have completed the work and successfully responded to my comments.